# Learning Multimodal Data Augmentation in Feature Space

**Zichang Liu** [*]
Department of Computer Science
Rice University
zl71@rice.edu

**Zhiqiang Tang**
Amazon Web Services
zqtang@amazon.com

**Xingjian Shi**
Amazon Web Services
xjshi@amazon.com

**Aston Zhang**
Amazon Web Services
astonz@amazon.com

**Mu Li**
Amazon Web Services
mli@amazon.com

**Anshumali Shrivastava**
Department of Computer Science
Rice University
anshumali@rice.edu

**Andrew Gordon Wilson**
New York University
Amazon Web Services
andrewgw@cims.nyu.edu

## Abstract

The ability to jointly learn from multiple modalities, such as text, audio, and visual data, is a defining feature of intelligent systems. While there have been promising advances in designing neural networks to harness multimodal data, the enormous success of data augmentation currently remains limited to single-modality tasks like image classification. Indeed, it is particularly difficult to augment each modality while preserving the overall semantic structure of the data; for example, a caption may no longer be a good description of an image after standard augmentations have been applied, such as translation. Moreover, it is challenging to specify reasonable transformations that are not tailored to a particular modality. In this paper, we introduce LeMDA, *Learning Multimodal Data Augmentation*, an easy-to-use method that automatically learns to jointly augment multimodal data in feature space, with no constraints on the identities of the modalities or the relationship between modalities. We show that LeMDA can (1) profoundly improve the performance of multimodal deep learning architectures, (2) apply to combinations of modalities that have not been previously considered, and (3) achieve state-of-the-art results on a wide range of applications comprised of image, text, and tabular data.

## 1 Introduction

Imagine watching a film with no sound, or subtitles. Our ability to learn is greatly enhanced through jointly processing multiple data modalities, such as visual stimuli, language, and audio. These information sources are often so entangled that it would be near impossible to learn from only one modality in isolation — a significant constraint on traditional machine learning approaches. Accordingly, there have been substantial research efforts in recent years on developing multimodal deep learning to jointly process and interpret information from different modalities at once (Baltrušaitis et al., 2017). Researchers studied multimodal deep learning from various perspectives such as model architectures (Kim et al., 2021b; Pérez-Rúa et al., 2019; Nagrani et al., 2021; Choi & Lee, 2019), training techniques (Li et al., 2021; Chen et al., 2019a), and theoretical analysis (Huang et al., 2021; Sun et al., 2020b). However, data augmentation for multimodal learning remains relatively unexplored (Kim et al., 2021a), despite its enormous practical impact in single modality settings.

---

[*]Work done while interning at Amazon Web Services.

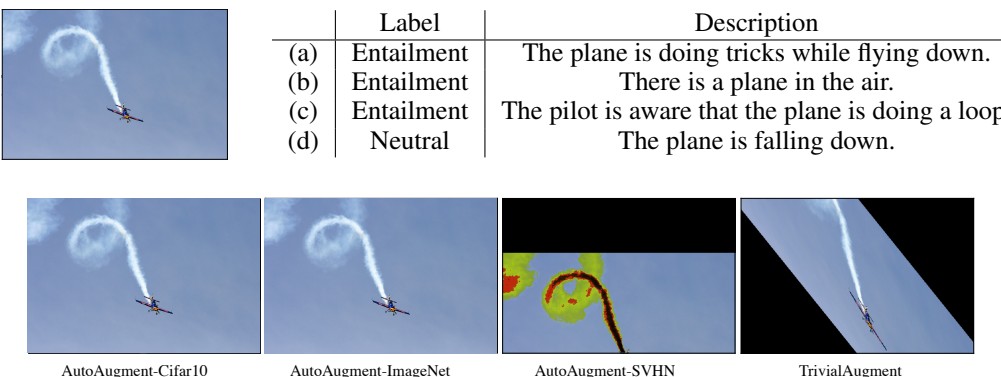

| | Label | Description |
|---|---|---|
| (a) | Entailment | The plane is doing tricks while flying down. |
| (b) | Entailment | There is a plane in the air. |
| (c) | Entailment | The pilot is aware that the plane is doing a loop. |
| (d) | Neutral | The plane is falling down. |

| AutoAugment-Cifar10 | AutoAugment-ImageNet | AutoAugment-SVHN | TrivialAugment |

Figure 1: The top row shows four training samples drawn from SNLI-VE (Xie et al., 2019a), a visual entailment dataset. Each text description is paired with the image on the top left. The task is to predict the relationship between the image and the text description, which can be "Entailment", "Neutral", or "Contradiction". The bottom row shows four augmented images generated by different image-only augmentation methods. If we pair the text description with the augmented images, we observe mislabeled data. For example, the smoke loop is cropped out in the image augmented via TrivialAugment. The new image does not match the description: "The pilot is aware that the plane is doing a loop", as in data (c). However, the label of the augmented pair will still be "Entailment".

Indeed, data augmentation has particularly proven its value for data efficiency, regularization, and improved performance in computer vision (Ho et al., 2019; Cubuk et al., 2020; Müller & Hutter, 2021; Zhang et al., 2017; Yun et al., 2019) and natural language processing (Wei & Zou, 2019; Karimi et al., 2021; Fadaee et al., 2017; Sennrich et al., 2015; Wang & Yang, 2015; Andreas, 2020; Kobayashi, 2018). These augmentation methods are largely tailored to a particular modality in isolation. For example, for object classification in vision, we know certain transformations such as translations or rotations should leave the class label unchanged. Similarly, in language, certain sentence manipulations like synonym replacement will leave the meaning unchanged.

The most immediate way of leveraging data augmentation in multimodal deep learning is to separately apply well-developed unimodal augmentation strategies to each corresponding modality. However, this approach can be problematic because transforming one modality in isolation may lead to disharmony with the others. Consider Figure 1, which provides four training examples from SNLI-VE (Xie et al., 2019a), a vision-language benchmark dataset. Each description is paired with the image on the top left, and the label refers to the relationship between the image and description. The bottom row provides four augmented images generated by state-of-the-art image augmentation methods (Cubuk et al., 2019; Müller & Hutter, 2021). In the image generated by AutoAugment-Cifar10 and AutoAugment-SVHN, the plane is entirely cropped out, which leads to mislabeling for data (a), (b), (c), and (d). In the image generated by AutoAugment-ImageNet, due to the change in smoke color, this plane could be on fire and falling down, which leads to mislabeling for data (a) and (d). In the image generated by TrivialAugment (Müller & Hutter, 2021), a recent image augmentation method that randomly chooses one transformation with a random magnitude, the loop is cropped out, which leads to mislabeling for data (a) and (c). Mislabeling can be especially problematic for over-parameterized neural networks, which tend to confidently fit mislabeled data, leading to poor performance (Pleiss et al., 2020).

There are two key challenges in designing a general approach to multimodal data augmentation. First, multimodal deep learning takes input from a diverse set of modalities. Augmentation transformations can be obvious for some modalities such as vision and language, but not others, such as sensory data which are often numeric or categorical. Second, multimodal deep learning includes a diverse set of tasks with different cross-modal relationships. Some datasets have redundant or totally correlated modalities while others have complementary modalities. There is no reasonable assumption that would generally preserve labels when augmenting modalities in isolation.

In this work, we propose LeMDA (*Learning Multimodal Data Augmentation*) as a general multimodal data augmentation method. LeMDA augments the latent representation and thus can be applied to any modalities. We design the augmentation transformation as a learnable module such that

it is adaptive to various multimodal tasks and cross-modal relationships. Our augmentation module is learned together with multimodal networks to produce informative data through adversarial training, while preserving semantic structure through consistency regularization. With no constraints over the modalities and tasks, one can simply plug-and-play LeMDA with different multimodal architectures. We summarize our contributions as follows.

- In Section 3, we introduce LeMDA, a novel approach to multimodal data augmentation. Section 3.1 shows how to use LeMDA with multimodal networks, and Section 3.2 describes how to train the augmentation module to produce informative and label preserving data. The method is notable for several reasons: (1) it can be applied to any modality combinations; (2) it is attractively simple and easy-to-use; (3) it is the first augmentation method to be applied to the joint of text, image, and tabular data, which is essentially uncharted territory.
- In Section 4, we show that LeMDA consistently boosts accuracy for multimodal deep learning architectures compared to a variety of baselines, including state-of-the-art input augmentation and feature augmentation methods.
- In Section 4.4, we provide an ablation study validating the design choices behind LeMDA. In particular, we study the architecture of the augmentation module, and the effects of consistency regularizer. We demonstrate that the consistency regularizer clearly outperforms $L_2$ regularization (Tang et al., 2020b).

## 2 BACKGROUND AND RELATED WORK

**Multimodal network architectures.** Multimodal deep learning architectures are categorized as performing early or late fusion, depending on the stage of combining information from each modality. In *early fusion*, the network combines the raw input or token embedding from all the modalities. Early fusion architectures can be designed to exploit the interaction between low-level features, making it a good choice for multimodal tasks with strong cross-modal correlations (Barnum et al., 2020; Gao et al., 2020). For example, there exist low-level correspondence in image captioning task because different words in the caption may relate to different objects in the image. We note that feature-space augmentation procedures are typically computationally intractable on early-fusion architectures, because early fusion would require combining a large number of latent features, such as a long sequence of token embeddings. On the other hand, in *late fusion*, the focus of our work, input from each modality is independently processed by different backbones. The representations provided by different backbones are fused together in later layers, often just before the classifier layer (Shi et al., 2021a; Wang et al., 2017; Schönfeld et al., 2019; Mahajan et al., 2020). This design is straightforward to apply to any new modality and any multimodal task. Late fusion often uses pre-trained networks as backbones in each modality, making it more computationally tractable. In both early and late fusion, there are a variety of methods to fuse information. Standard approaches include (1) feed all modalities as token embedding into the network, (2) perform cross-attention between modalities, (3) concatenate representations from all modalities, and (4) combine the predictions from each modality in an ensemble (Baltrušaitis et al., 2017). Researchers usually design the multimodal network by considering the task objective, the amount of data available, and the computation budget (Shi et al., 2021b; Chen et al., 2019b; Li et al., 2022; Tsai et al., 2019; Mahajan & Roth, 2020). Baltrušaitis et al. (2017) provides further readings.

**Data augmentation for single modality tasks.** Data augmentation is widely adopted in vision and natural language tasks. In vision, we can manually intervene on a per-task basis to apply transformations that should leave our label invariant — e.g., translations, rotations, flipping, cropping, and color adjustments. A transformation on one task may not be suitable for another: for example, flipping may be reasonable on CIFAR-10, but would lose semantic information on MNIST, because a flipped '6' becomes a '9'. Accordingly, there are a variety of works for automatic augmentations in vision, including neural architecture search (Ho et al., 2019; Cubuk et al., 2020), reinforcement learning (Cubuk et al., 2019), generative modelling (Ratner et al., 2017), mixing aspects of the existing data (Zhang et al., 2017; Yun et al., 2019), and adversarial training for informative examples (Fawzi et al., 2016; Goodfellow et al., 2015; Zhang et al., 2019; Suzuki, 2022; Tang et al., 2020b; Tsai et al., 2017) . Similarly, in natural language processing there are a variety of standard interventions (replacement, deletion, swapping) (Wei & Zou, 2019; Karimi et al., 2021; Fadaee et al., 2017),

---

**Algorithm 1** LeMDA Training

---

**Input:** Task network before fusion $\mathcal{F}_{\text{before}}$; Task network after fusion $\mathcal{F}_{\text{after}}$; Augmentation network $\mathcal{G}$; Training set $\mathcal{X}$; Task loss function $L$; Consistency loss $L_{\text{consist}}$;

**while** $\mathcal{F}$ not converged **do**

    Sample a mini-batch from $\mathcal{X}$

    Compute $z \leftarrow \mathcal{F}_{\text{before}}(x)$

    Generate augment feature $\mathcal{G}(z)$

    $\hat{y} \leftarrow \mathcal{F}_{\text{after}}(z), \hat{y}_{\mathcal{G}} \leftarrow \mathcal{F}_{\text{after}}(\mathcal{G}(z))$

    Update the augmentation network $\mathcal{G}$ by stochastic gradient $-\nabla L(\hat{y}_{\mathcal{G}}) + \nabla L_{\text{consist}}(\hat{y}, \hat{y}_{\mathcal{G}})$

    Update the task network $\mathcal{F}$ by stochastic gradient $\nabla L(\hat{y}) + \nabla L(\hat{y}_{\mathcal{G}})$

**end while**

---

and more automatic approaches such as back-translation (Sennrich et al., 2015), context augmentation (Wang & Yang, 2015; Andreas, 2020; Kobayashi, 2018), and linear interpolation of training data (Sun et al., 2020a). Data augmentation is less explored for tabular data, but techniques in vision, such as mixup (Zhang et al., 2017) and adversarial training (Goodfellow et al., 2015) have recently been adapted to the tabular setting with promising results (Kadra et al., 2021). Latent space augmentation is much less explored than input augmentation, as it is less obvious what transformations to apply. To augment latent vectors produced by passing data inputs through a neural network (*feature space* augmentation), researchers have considered interpolation, extrapolation, noise addition, and generative models (Verma et al., 2019; Liu et al., 2018; Kumar et al., 2019).

**Multimodal data augmentation.** There are a small number of works considering multimodal data augmentation, primarily focusing on vision-text tasks. In visual question answering, Tang et al. (2020a) proposes to generate semantic similar data by applying back-translation on the text and adversarial noise on the image. Wang et al. (2021) generates text based on images using a variational autoencoder. In cross-modal retrieval, Gur et al. (2021) query similar data from external knowledge sources for cross-modal retrieval tasks. The state-of-the-art augmentation procedure for visual-language representation learning generates new image-text pairs by interpolating between images and concatenating texts in a method called *MixGen* (Hao et al., 2022).

All prior work on multimodal data augmentation relies on tailored modality-specific transformations. By contrast, our proposed approach is fully automatic and can be applied to any arbitrary modality. Indeed, for the first time, we consider augmentation jointly over the tabular, image, and language modalities. Moreover, even for image-text specific problems, we show that our approach outperforms MixGen, the state-of-the-art tailored approach.

## 3    LeMDA: Learning Multimodal Data Augmentation

We now introduce LeMDA, a simple and automatic approach to multi-modal data augmentation. LeMDA learns an *augmentation network* $\mathcal{G}$, along with the multimodal *task network* $\mathcal{F}$ to generate informative data that preserves semantic structure. In Sections 3.1 and 3.2 we describe how we learn the parameters the augmentation and task networks, respectively. We summarize the training algorithm for LeMDA in Figure 2 and Algorithm 1. In Section 3.4 we provide intuition for the consistency loss. Finally, in Section 3.3 we describe how we design the augmentation network.

### 3.1    Training the Task Network

The task network can be divided into two parts at the fusion layer $\mathcal{F}(x) = \mathcal{F}_{\text{after}}(\mathcal{F}_{\text{before}}(x))$ where $\mathcal{F}_{\text{before}}$ denotes the layers before fusion, $\mathcal{F}_{\text{after}}$ denotes the layers after the fusion. Given a training sample $x$, we pass $x$ until the fusion layer and obtain the latent features for each modality $\{z_i\}_{i=1}^N = \mathcal{F}_{\text{before}}(x)$ where $N$ is the number of modalities. Taken $\{z_i\}_{i=1}^N$ as inputs, the augmentation network $\mathcal{G}$ generates additional latent vectors $\mathcal{G}(\{z_i\}_{i=1}^N)$. Both $\{z_i\}_{i=1}^N$ and $\mathcal{G}(\{z_i\}_{i=1}^N)$ are fed through the rest of target network $\mathcal{F}_{\text{after}}$ as distinct training data. Then, the task network is trained in the standard way, taking the task loss function on both original data and augmented data, to find $\min \mathbb{E}_{x \sim \mathcal{X}}(L(\hat{y}) + L(\hat{y}_{\mathcal{G}}))$ where $\hat{y} = \mathcal{F}_{\text{after}}(\mathcal{F}_{\text{before}}(x))$ and $\hat{y}_{\mathcal{G}} = \mathcal{F}_{\text{after}}(\mathcal{G}(\mathcal{F}_{\text{before}}(x)))$.

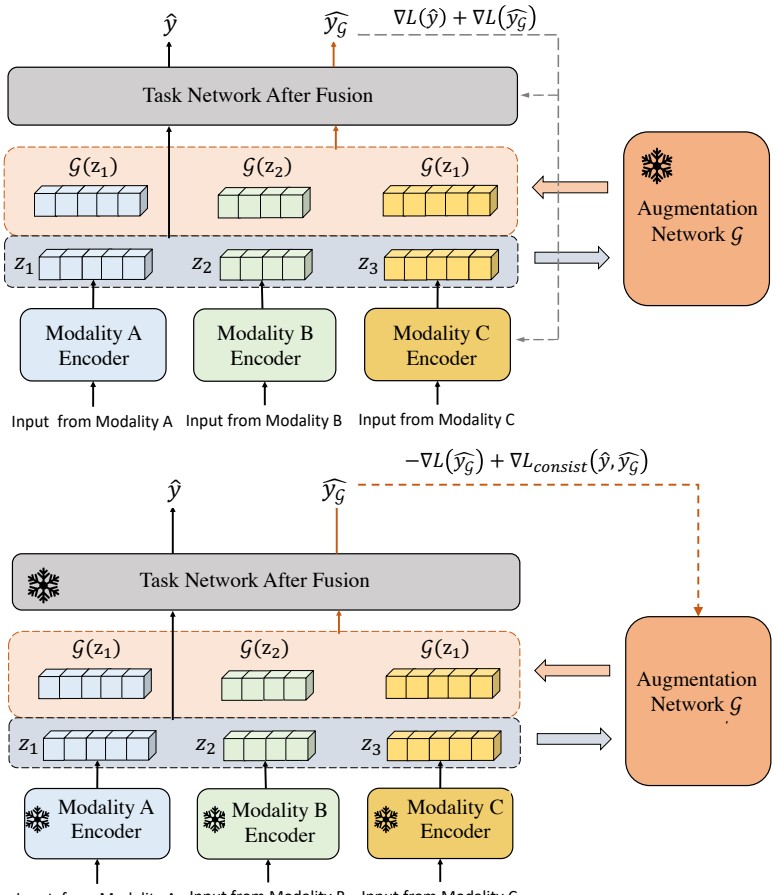

Figure 2: LeMDA training as described in Algorithm 1. **Top:** the training process for the task network. Latent representations for each modality $z_i$ are passed into the augmentation network, which generates a new latent vector for each modality. Both original features and augmented features are passed into the rest of the task network. **Bottom:** the training process for the augmentation network. The augmentation network is trained to maximize task loss while minimizing consistency loss. We describe our standard choices for fusion in Section 2, and the design of our augmentation network in Section 3.3.

## 3.2 TRAINING THE AUGMENTATION NETWORK

Inspired by adversarial data augmentation, we optimize parameters for the augmentation network to maximize the task loss such that the task network's representation is encouraged to be updated by the augmented data. At the same time, we introduce a consistency regularizer that encourages a similar output distribution given the original data and the augmented data to preserve the semantic structure. Formally, we find $\max \mathbb{E}_{x \sim \mathcal{X}}(L(\hat{y}_{\mathcal{G}})) + \min \mathbb{E}_{x \sim \mathcal{X}}(L_{\text{consist}}(\hat{y}, \hat{y}_{\mathcal{G}}))$ where $L_{\text{consist}}(\hat{y}, \hat{y}_{\mathcal{G}})$ denotes a divergence metric between the logit outputs on original data $\hat{y}$ and on augmented data $\hat{y}_{\mathcal{G}}$ such as the Kullback-Leibler divergence.

**Confidence masking.** For classification problems, we apply the consistency term only to the samples whose highest probability is greater than a threshold $\alpha$. If the task network can't make a confident prediction, it is unlikely the prediction provides a good reference to the ground truth label.

**Design decisions.** The simplicity and generality of this approach, combined with its strong empirical performance in Section 4, are LeMDA's most appealing features. The few design decisions for training involve how the consistency regularizer should be defined and to what extent it should be applied. For example, as an alternative to a KL-based consistency regularizer, we could minimize the $L_2$ distance of the augmented feature vector to the original feature vector as a proxy for preserving the label of the augmentation. We provide ablations of these factors in Section 4.4.

### 3.3 The Design of Augmentation Network

The augmentation network can take various forms depending on the multimodal learning tasks and the fusion strategies. In our experiments, we use a variational autoencoder (VAE) as the augmentation network, since VAEs have generally been found effective for augmentation purposes (Tang et al., 2020b). We consider two architectural choices:

**MLP-VAE:** The encoder and decoder of VAE are MLPs. $\{z_i\}_{i=1}^N$ are concatenated as the input.

**Attention-VAE:** The encoder and decoder are made of self-attention and feedforward networks. $\{z_i\}_{i=1}^N$ are treated as $N$ tokens where each token has an embedding $z_i$.

There are two loss terms in the standard VAE, the reconstruction loss, and the KL divergence regularizer. We only adopt the KL regularizer on the encoder distribution. The updating step for augmentation networks is $-\nabla L(\hat{y}_\mathcal{G}) + \nabla L_{\mathsf{consist}}(\hat{y}, \hat{y}_\mathcal{G}) + +\nabla L_{\mathsf{VAE}}$, where $L_{\mathsf{VAE}}$ refers to the KL divergence regularizer on the latent encoding distribution.

The major deciding factor between MLP-VAE and Attention-VAE is the multimodal task network architectures. With late fusion architectures, which is the primary focus of this paper, $z_i$ refers to the representation from a single modality backbone (e.g. CLS embedding from a BERT model), and $N$ is the number of modalities or the number of backbone models. We can concatenate $\{z_i\}_{i=1}^N$ as one vector input to MLP-VAE, or we can treat $\{z_i\}_{i=1}^N$ as a sequence of $N$ tokens to Attention-VAE. Attention-VAE may be less intuitive here because $N$ is usually a small number in late fusion architectures( 2 or 3 in our experiment). We provide a performance comparison between these two architectures in Section 4.4. On the other hand, for early fusion architectures, $z_i$ could be a sequence of token embedding for a text or a sequence of patch embedding for an image. Concatenation will result in a really high dimension input, which makes MLP-VAE less favorable.

### 3.4 Intuition on Why Consistency Regularizer Discourages Mislabeled Data

In Figure 3 we provide intuition for the consistency regularizer using a simple illustrative binary classification. Darker background corresponds to higher task training loss, the solid green line is the actual decision boundary, and the dashed green line is the model's decision boundary. Starting from a point in feature space, moving to D1 and D2 would provide a similar increase in task loss and thus are equally favored by the adversarial loss term. However, D2 crosses the model's decision boundary, and thus would be heavily penalized by the consistency regularizer — as we would hope, since such a point is likely to have a different class label. On the other hand, an L2 regularizer between the original and augmented points in feature space would have no preference between D1 and D2, as they are an equal distance away from the starting point. Empirically, in Section 4, we see the consistency loss confers accuracy improvements over both pure adversarial training and L2 regularizer.

Similar intuition is in Suzuki (2022), which uses the logits distribution from the teacher model (an exponential moving average over the model's weights) as the soft target such that the augmented data is still recognizable by the teacher, and Xie et al. (2019b), which designs the unsupervised training objective to encourage similar logits for augmented data.

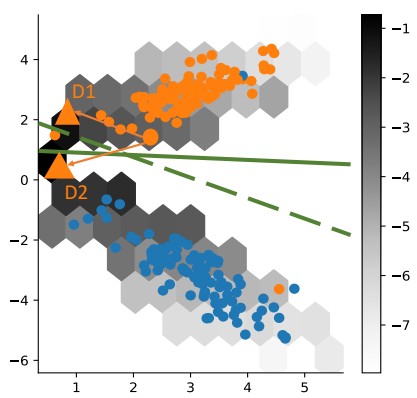

Figure 3: Motivation for the consistency regularizer. The solid and dashed green lines are the ground truth and model decision boundaries, respectively. Darker background corresponds to a higher loss for the task network. We intuitively prefer D1, because the augmented point should be informative but preserve the same label. The consistency loss will prefer D1 over D2, because D2 crosses the model's decision boundary, even though both points incur the same training loss.

| Dataset | # Train | #Test | Metric | Image | Text | Tabular |
|---------|---------|-------|--------|-------|------|---------|
| Hateful Memes | 7134 | 1784 | Accuracy | ✓ | ✓ | |
| Food101 | 67972 | 22716 | Accuracy | ✓ | ✓ | |
| SNLI-VE | 529527 | 17901 | Accuracy | ✓ | ✓ | |
| Petfinder | 11994 | 2999 | Quadratic Kappa | ✓ | ✓ | ✓ |
| Melbourne Airbnb | 18316 | 4579 | Accuracy | | ✓ | ✓ |
| News Channel | 20284 | 5071 | Accuracy | | ✓ | ✓ |
| Wine Reviews | 84123 | 21031 | Accuracy | | ✓ | ✓ |
| Kick Starter Funding | 86502 | 21626 | ROC-AUC | | ✓ | ✓ |

Table 1: This table provides a summary of the source, statistic, and modality identity.

## 4 EXPERIMENTS

We evaluate LeMDA over a diverse set of real-world multimodal datasets. We curate a list of public datasets covering image, text, numerical, and categorical inputs. Table 1 provides a summary of the source, statistic, and modality identity. We introduce baselines in Section 4.1, and describe experimental settings in Section 4.2 We provide the main evaluation result in Section 4.3. Finally, we investigate the effects of the consistency regularizer and the choices of augmentation model architecture in Section 4.4.[1]

### 4.1 BASELINES

To the best of our knowledge, there exist no general-purpose multimodal augmentation methods. We compare against a diverse set of state-of-the-art data augmentation methods from vision, language, and vision-text tasks. We additionally consider baselines for feature augmentation, since LeMDA augments in the feature space. Finally, we compare with state-of-the-art multimodal augmentation methods from the vision-text tasks, although we note, unlike LeMDA, these methods are not general purpose and cannot be directly applied to our datasets that have tabular inputs.

- **Input Augmentation.** We apply state-of-the-art input augmentation independently on the data from each modality. For images, we use TrivialAugment (Müller & Hutter, 2021), a simple and effective method for image classification tasks. For text, we apply EDA (Wei & Zou, 2019) and AEDA (Karimi et al., 2021). We randomly sample one transformation from all transformations proposed in EDA and AEDA with a randomly generated magnitude.

- **Mixup.** Mixup was originally proposed to perform interpolation between two images in the training data for image classification. We adopt the original Mixup for images and numerical features and extend it for text and categorical features. Specifically, given a pair of data, we construct the mixed data as follows. We generate a random number $j$ uniformly between 0.0 to 1.0. If $j < \alpha$, we use the first data, else we use the second data.

- **Manifold Mixup.** Manifold Mixup (Verma et al., 2019) performs interpolation between hidden representations and thus can be applied to all modalities. We applied Manifold Mixup to the exact feature in the multimodal network as LeMDA.

- **MixGen.** MixGen (Hao et al., 2022) is a state-of-the-art data augmentation designed specifically for vision-text tasks. MixGen generates new data by interpolating images and concatenating text. We apply MixGen to datasets only consisting of images and text.

### 4.2 EXPERIMENT SETUP

We use Multimodal-Net (Shi et al., 2021a) for all the datasets except SNLI-VE. Multimodal-Net passes input from each modality through separate backbones, concatenates the representation(e.g. the CLS embedding) from all backbones, and passes them through fusion MLP. We use the default hyper-parameters provided by Multimodal-Net and plug LeMDA before the fusion layer. We use ConvNet as the image backbone and ELECTRA as the text backbone.

To further demonstrate LeMDA's generalizability, we evaluate LeMDA with early fusion architectures ALBEF (Li et al., 2021) on SNLI-VE. ALBEF performs cross-attention between image patch

---

[1]Code is available at https://github.com/lzcemma/LeMDA/

|  | Multimodal Network | Input Augmentation | Mixup | Manifold MixUp | MixGen | LeMDA |
|---|---|---|---|---|---|---|
| Hateful Memes | 0.6939 | 0.7057 | 0.6939 | 0.6878 | 0.7510 | **0.7562** |
| Food101 | 0.9387 | 0.9432 | 0.9400 | 0.9390 | 0.9432 | **0.9452** |
| Petfinder | 0.2911 | 0.3236 | 0.3244 | 0.3492 | - | **0.3537** |
| Melbourne Airbnb | 0.3946 | 0.3978 | 0.3966 | 0.3840 | - | **0.4047** |
| News Channel | 0.4754 | 0.4745 | 0.4723 | 0.4757 | - | **0.4798** |
| Wine Reviews | 0.8192 | 0.8212 | 0.8143 | 0.8126 | - | **0.8262** |
| Kick Starter Funding | 0.7571 | 0.7572 | 0.7597 | 0.7578 | - | **0.7614** |
| SNLI-VE | 0.7916 | 0.7931 | 0.7957 | 0.7929 | 0.7950 | **0.7981** |

Table 2: LeMDA not only significantly increases accuracy over the original architectures but also outperforms all baselines.

embeddings and text token embeddings. We keep all original configurations except the batch size due to limitations in computation memory. We set the batch size to be half of the default. We load the 4M pre-trained checkpoints. In this setting, we apply LeMDA before the cross-attention layer. The augmentation network augments every image patch embedding and every text token embedding.

For LeMDA, we set the confidence threshold for consistency regularizer $\alpha$ as 0.5, and we study this choice in Section 4.4. For our baselines, we follow the recommended hyperparameters. For Mixup and Manifold Mixup, we set $\alpha$ as 0.8, and for MixGen, we set $\lambda$ as 0.5.

## 4.3 MAIN RESULTS

We summarize the performance comparison in Table 2. Plugging LeMDA in both Multimodal-Net and ALBEF leads to consistent accuracy improvements. There are also some particularly notable improvements, such as a 6% increase in accuracy for both Hateful Memes and Petfinder. Table 2 illustrates how LeMDA performs comparing to the baselines. We see that single modality input augmentation methods can hurt accuracy, for example, on News Channel, in accordance with the intuition from our introductory example in Figure 1. Mixup also can hurt accuracy, for example, on Wine Reviews. Similarly, in the latent space, Manifold Mixup fails to improve accuracy across datasets. On Melbourne Airbnb and Wine Reviews, Manifold Mixup results in accuracy drops. On the contrary, LeMDA consistently improves upon original architectures and provides clearly better performance than a wide range of baselines.

## 4.4 ABLATION STUDY

We now perform three ablation studies to support the design choice of LeMDA.

| Dataset | No Regularizer | Consistency | L2 | Consistency + L2 |
|---|---|---|---|---|
| Hateful Memes | 0.7433 | **0.7562** | 0.7472 | 0.7545 |
| Food101 | 0.9433 | **0.9452** | 0.9415 | 0.9438 |
| Petfinder | 0.3369 | **0.3537** | 0.3420 | 0.3461 |
| Melbourne Airbnb | 0.3935 | 0.4047 | 0.3987 | **0.4051** |
| News Channel | 0.4851 | 0.4869 | 0.4869 | **0.4894** |
| Wine Reviews | 0.8228 | **0.8263** | 0.8255 | 0.8262 |
| Kick Starter Funding | 0.7609 | **0.7614** | 0.7604 | **0.7614** |

Table 3: The effects of regularizer choice. Regularization over the augmentation network generally lead to better performance. Consistency regularizer consistently outperforms a standard L2 regularizer in feature space. Moreover, combining the consistency regularizer with an L2 regularizer improves over only using an L2 regularizer.

**Augmentation Network Regularizer.** We argue in Section 3.4 that consistency regularizer helps preserve the semantic structure of augmentations. In Table 3, we see that this consistency regularizer significantly improves performance, and also outperforms L2 regularization in feature space. While L2 regularization attempts to keep augmented features close in distance to the original as a proxy for semantic similarity, the consistency regularization has access to the softmax outputs of the target and augmentation networks, providing direct information about labels.

| Dataset | No Augmentation | MLP-VAE | Attention-VAE |
|---|---|---|---|
| Hateful Memes | 0.6939 | **0.7562** | 0.7483 |
| Food101 | 0.9387 | **0.9452** | 0.9443 |
| Petfinder | 0.2911 | **0.3537** | 0.3456 |
| Melbourne Airbnb | 0.3946 | **0.4047** | 0.4031 |
| News Channel | 0.4754 | **0.4798** | 0.4733 |
| Wine Reviews | 0.8192 | **0.8262** | 0.8250 |
| Kick Starter Funding | 0.7571 | **0.7614** | 0.7586 |

Table 4: Both MLP-VAE and Attention-VAE augmentation networks provide significant gains over no augmentation. MLP-VAE outperforms Attention-VAE in the late fusion setting because the input of augmentation networks is only 2 or 3 latent representations.

| Dataset | $\alpha = 0$ | $\alpha = 0.3$ | $\alpha = 0.5$ | $\alpha = 0.8$ |
|---|---|---|---|---|
| Hateful Memes | 0.7410 | 0.7443 | **0.7556** | 0.7371 |
| Food101 | 0.9431 | 0.9438 | **0.9447** | 0.9438 |
| Petfinder | 0.3243 | 0.3462 | **0.3676** | 0.3497 |
| Melbourne Airbnb | 0.3988 | 0.3964 | **0.3988** | 0.3964 |
| News Channel | **0.4869** | 0.4851 | **0.4869** | 0.4695 |
| Wine Reviews | 0.8228 | 0.8274 | **0.8275** | 0.8274 |
| Kick Starter Funding | 0.7614 | 0.7617 | **0.7620** | 0.7618 |

Table 5: The influence of confidence-based masking. $\alpha = 0$ indicates no masking such that consistency loss is calculated with all data. We see that filtering out low-confidence data leads to better end-to-end accuracy.

**Architecture Difference.** We consider the two augmentation architectures introduced in Section 3.3, MLP-VAE and Attention-VAE. In Table 4 we see both architectures increase performance over no augmentation. We also see that MLP-VAE generally outperforms Attention-VAE. We suspect the reason is that Multimodal-Net passes the concatenation of $N$ latent vector into fusion layers, where $N$ is the number of modalities (2 or 3 in our experiments). For Attention-VAE, this means that the input is only 2 or 3 tokens. However, we note that MLP-VAE is not reasonable for ALBEF, since it would require concatenating thousands of tokens.

**Confidence Masking.** Here, we investigate the effect of confidence masking, as well as the choice of $\alpha$ in Table 5. $\alpha = 0$ means no masking, and all training data are used to calculate consistency loss. We see that confidence masking generally leads to higher accuracy, and that the performance is not particularly sensitive to the precise value of $\alpha$.

## 4.5 The Relationship between Modalities

We can categorize the relationship between available modalities by looking at $P(y|x)$ where $y \sim \mathcal{Y}$ and $\mathcal{Y}$ is the target domain. Let $x = \{x_1, x_2, \ldots, x_N\}$ consist of $N$ modalities.

**Perfect Correlation** $P(y|x) = P(y|x_n)$. Essentially, one modality alone provides enough information to make the right prediction. Nevertheless, data still comprises multiple modalities for reasons such as easier training (Huang et al., 2021). One example could be Food101, where the task is to predict the food from the text of a recipe and the photo of the food.

**Complementary** $P(y|x) = P(y|\{x_1, x_2, \ldots, x_N\})$. This category suggests that information aggregated from all modalities is necessary to make the right prediction. Each modality is complementary to each other, and missing one modality would lead to information loss. One example could be Hateful Memes, where only the combined meaning of text and image indicates harmful content.

The design for LeMDA does not exploit any assumption over the cross-modal relationship. We observe from Table 2 that LeMDA consistently improves performance regardless of the relationship.

## 5 Conclusion

Jointly learning from multiple different modalities will be crucial in our quest to build autonomous intelligent agents. We introduce the first method, LemDA, for jointly learning data augmentation

across arbitrary modalities. LeMDA is simple, automatic, and achieves promising results over a wide range of experiments. Moreover, our results provide several significant conceptual findings about multimodal data augmentation in general: (1) separately augmenting each modality performs much worse than joint augmentation; (2) although feature augmentation is less popular than input augmentation for single-modality tasks because it is less interpretable, feature augmentation is particularly promising for modality-agnostic settings; (3) a learning-based multimodal augmentation policy can outperform even tailored augmentations, and significantly improve accuracy when augmentation transformations are not obvious such as for categorical data.

Our investigation has primarily focused on late-fusion architectures, showing strong results over a wide range of settings. In general, applying feature augmentation strategies to early-fusion architectures is an open question. Early fusion combines a large number of latent features (e.g., a long sequence of token embeddings), resulting in typically intractable computational costs for augmenting every latent feature. Our experiment with an early-fusion architecture shows however that developing more efficient augmentation networks, or selectively generating only a few important latent vectors, is a promising direction for future work.

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

## A    MORE DETAILS ON THE DESIGN

### A.1    DETAILED ARCHITECTURES OF THE AUGMENTATION NETWORK

We consider two VAE architectures in LeMDA, depending on the architecture of the task network. The latent dimension in VAE is set as 8. We adopted the KL divergence regularizer on the encoder distribution. Note that we do not use the reconstruction loss between the input and the output. In MLP-VAE, the encoder and decoder are standard fully connected layers with ReLU as the activation function. Dropout is used with $p = 0.5$. In Attention-VAE, the encoder is implemented as torch.nn.TransformerEncoder. We set numlayers as 4 and nhead as 8. One fully connected layer is used to mapThe decoder is symmetric as the encoder. Features from all modalities are treated as token embedding with no cross-attention.

We use VAE for its simplicity. The main focus of this paper is to demonstrate the effectiveness of a learnable augmentation network for multimodal learning. Other generative models, such as diffusion models and GANs, are also valid architectures. The main concern may lie in efficiency, and we leave this direction as future work.

### A.2    IMPLEMENTATION DETAILS OVER THE TRAINING PROCEDURE

In practice, we iterative train the task and augmentation networks using the same batch of training data. Specifically, we perform two separate forward passes using $\mathcal{F}_{\text{after}}$ for easy implementation with pyTorch Autograd. We use two optimizers, one for the task network and one for the augmentation network.

## B    EXPERIMENT DETAILS

### B.1    ADDITIONAL STUDIES ON THE TRAINING COST

One limitation of a learning-based approach is the extra training cost. LeMDA optimizes the augmentation network along with the task network and does incur extra training costs. Here, we investigate the training throughput to provide a more complete understanding of the method. We summarize the training throughput(it/second) in Table 6. As expected, we observe lower throughput for LeMDA compared to other baselines.

|  | Multimodal Network | Input Augmentation | Mixup | Manifold MixUp | MixGen | LeMDA |
|---|---|---|---|---|---|---|
| Hateful Memes | 2.39 | 2.17 | 2.35 | 1.63 | 2.35 | 1.41 |
| Food101 | 4.27 | 4.46 | 4.31 | 4.48 | 4.47 | 2.21 |
| Petfinder | 2.36 | 2.29 | 2.95 | 2.36 | - | 1.87 |
| Melbourne Airbnb | 5.66 | 5.94 | 5.59 | 5.69 | - | 4.13 |
| News Channel | 8.14 | 7.18 | 7.31 | 7.12 | - | 5.12 |
| Wine Reviews | 12.54 | 11.60 | 11.89 | 11.46 | - | 6.28 |
| Kick Starter Funding | 12.37 | 12.57 | 12.62 | 12.21 | - | 6.69 |

Table 6: This table summarizes the training throughput, measured as it/second. Experiments were conducted on a server with 8 V100 GPU. As expected, learning-based approach incur higher training cost.

However, efficiency can be improved. The most straightforward direction is to reduce the frequency of updating the augmentation network. Currently, the augmentation network is updated every iteration. However, the parameters for our task network change slowly, especially in the later stage of training. We leave this part as future direction.

The optimization for the augmentation network is a min-max game, which leads to hyperparameters to balance the contradicting loss. Specifically, $-w_1 \nabla L(\hat{y}_{\mathcal{G}}) + w_2 \nabla L_{\text{consist}}(\hat{y}, \hat{y}_{\mathcal{G}}) + w_3 \nabla L_{\text{VAE}}$, where $L_{\text{VAE}}$ refers to the KL divergence regularizer on the latent encoding distribution.

In our main experiment, we use $w_1 = 0.0001, w_2 = 0.1, w_3 = 0.1$ on all datasets except Melbourne Airbnb and SNLI-VE. On Melbourne Airbnb and SNLI-VE, we use $w_1 = 0.001, w_2 = 0.1, w_3 = 0.1$. Note that the hyperparameters are relative consistent across datasets.

Further, we investigate the influence of the different combinations of $w_1$, $w_2$, and $w_3$. We summarize the result on Petfinder Table 7. We observe consistent improvements over the original multimodal network across various combinations.

| $w_1$ | $w_2$ | $w_3$ | Accuracy |
|---|---|---|---|
| 0.0001 | 0.1 | 0.1 | 0.3539 |
| 0.0001 | 0.01 | 0.01 | 0.3400 |
| 0.005 | 0.1 | 0.1 | 0.3482 |
| 0.005 | 0.01 | 0.01 | 0.3464 |
| 0.001 | 0.1 | 0.1 | 0.3371 |
| 0.001 | 0.01 | 0.01 | 0.3467 |
| Multimodal Network | | | 0.2911 |

Table 7: LeMDA improves over Multimodal Network in accuracy with different sets of hyperparameters on Petfinder. Specifically, $-w_1 \nabla L(\hat{y}_{\mathcal{G}}) + w_2 \nabla L_{\text{consist}}(\hat{y}, \hat{y}_{\mathcal{G}}) + w_3 \nabla L_{\text{VAE}}$

## C MOTIVATIONAL EXAMPLES WHEN ONLY AUGMENTING ONE MODALITY

We have performed an additional set of experiments to investigate the effect of augmenting a single modality on Hateful Memes using state-of-the-art augmentation techniques. In Hateful Memes, both text and image are required to decide if the content is hateful. Two modalities provide complementary information to each other. We run baseline augmentation to only one modality or independently on two modalities. We observe no consistent improvements. Essentially, performing augmentation naively to one modality or jointly without considering cross-modality relationships won't lead to effective augmentation.

| Multimodal Network | Method | Image | Text | Image + Text |
|---|---|---|---|---|
| | Trivial Augment | 0.7040 | 0.6860 | 0.7057 |
| 0.6939 | MixUp | 0.6855 | 0.6777 | 0.6939 |
| | Manifold Mixup | 0.6323 | 0.7444 | 0.6878 |
| | MixGen | 0.7427 | 0.6872 | 0.7510 |

