# OpenReview forum: "Learning Multimodal Data Augmentation in Feature Space"
_ICLR.cc/2023/Conference — ICLR 2023 poster_

### Official Review · Reviewer_nNn5 · 2022-10-23

**Confidence:** 3
**Correctness:** 3
**Technical Novelty And Significance:** 3
**Empirical Novelty And Significance:** 3
**Recommendation:** 6

**Clarity, Quality, Novelty And Reproducibility:**

Quality: This is a timely work, proposing a data augmentation method under multi-modality setting. The method works well empirically.

Clarity: The paper generally flows well, except for some major confusing/missing points.

Novelty: Although the idea of training a separate NN to do data augmentation is not new, the multimodality setting is comparably under-explored yet.

Reproducibility: The authors promise they will release the code.


**Strength And Weaknesses:**

The paper flows well. But I am confused at some major settings. First, how $\mathcal{F}$ and $\mathcal{G}$ is trained. For the optimization problem definition under section 3.2, do you mean this (as I inferred from the context),

max_$\mathcal{G}$ $\mathbb{E}$\_{$x\sim\mathcal{X}$} (L($\hat{y}$ \_{$\mathcal{G}$})+L\_{consist}($\hat{y}$, $\hat{y}_{\mathcal{G}}$)).

(sorry there are some formatting issue from OpenReview) I am asking because I do not see why $L(\hat{y})$ is dependent on $\mathcal{G}$, and why this is a maxmin problem. If so, the notations in the algorithm and the figure are also wrong.

Also, for the VAE implementation, do you only adopt the architecture of VAE or both the architecture and VAE loss?

In practice, you do not have a coefficient to balance different losses?

In practice, how do you balance the training of the task network and augmentation network? For example, do you iteratively train either of the networks like GAN?

The experiment results look promising. The authors also provide comprehensive ablation study.

As an important motivation, it would be better if the claims on the relationships are verified via experiments. For example, what if in other methods/in the proposed method, only one of the complementary modalities is augmented?


**Summary Of The Paper:**

This work proposes LeMDA (Learning Multimodal Data Augmentation) as a general multimodal data augmentation method. Specifically, the authors design the augmentation transformation as a learnable module applying on the latent representations of the inputs from different modalities. Since the module is end-to-end learned, it is naturally adaptive to different cross-modal relationships and can be applied to any modality combinations. The empirical experiments show that the proposed method consistently boosts accuracy.

**Summary Of The Review:**

Overall, this is a timely paper with good empirical performance. However, I am not recommending acceptance because I am not sure I am understanding the main method correctly. Please do elaborate in the rebuttal on my questions above.

---

> ### Author Response · Authors · 2022-11-17
> **Response to Reviewer nNn5**
>
> Thank you for your supportive feedback! Inspired by your comments, we have performed several additional experiments. We hope you can consider raising your score in light of our response, and our general comment, highlighting the timeliness and significance of this work. Not only is the approach simple and effective over many problems, but we are also considering entirely uncharted territory, for example, with text+tabular+image modalities.
>
>
>
> **Q1: Clarification on the optimization definition in Section 3.2.**
>
> We encourage the augmentation network to maximize task loss to generate hard examples while minimizing consistency loss to preserve semantic meaning. We made a typo in the formulation, as $L(\hat{y})$ shouldn’t depend on G. A clearer definition could be
> $ \max\mathbb{E}\_{x \sim \mathcal{X}}(L(\hat{y}\_{\mathcal{G}})) + \min\mathbb{E}\_{x \sim \mathcal{X}}(L\_{\mathsf{consist}}(\hat{y}, \hat{y}\_{\mathcal{G}}) ) $. We updated the manuscript on the definition in Section 3.2. Thank you for the question.
>
>
>
> **Q2: For the VAE implementation, do you only adopt the architecture of VAE or both the architecture and VAE loss?**
>
> There are two loss terms in the standard VAE, the reconstruction loss, and the KL regularizer. We only adopt the KL divergence regularizer on the encoder distribution. We have now clarified this point in the manuscript.
>
>
> **Q3: Do you have a coefficient to balance different losses?**
>
>
>
> Since the optimization for the augmentation network is a min-max game, there are hyperparameters to balance the loss. Specifically, $w\_1 \partial L(\hat{y\_G}) + w\_2\partial L\_{consistent} +w\_3 \partial L\_{KL}$, where $\partial L\_{KL}$ is the VAE KL regularizer.
>
> We note that, relatively speaking, our method does not involve many hyperparameters and that performance is not particularly sensitive to the values of these hyperparameters. For example,
> in the following table, we summarize the hyperparameters we use for Table 1. The hyperparameters are pretty consistent across datasets.
>
>
> |  | w1 | w2| w3|
> |--|--|--|--|
> | Hateful Memes | 0.0001 | 0.1 |0.1 |
> | Food101| 0.0001 | 0.1 |0.1 |
> | Petfinder | 0.0001 | 0.1 |0.1 |
> | Melbourn Airbnb | 0.001 | 0.1 |0.1 |
> | New Channel | 0.0001 | 0.1 |0.1 |
> | Wine Review | 0.0001 | 0.1 |0.1 |
> | Kick Starter FUnding | 0.0001 | 0.1 |0.1 |
> | SNLI-VE | 0.001 | 0.1 |0.1 |
>
>
>
> Additionally, we investigate the influence of the different combinations of hyperparameters and summarize the accuracy in the following table on Petfinder. We observe a consistent improvement across various combinations.
>
> | Multimodel Network | | | 0.2911|
> |--|--|--|--|
> | w1 | w2 | w3| Accuracy|
> | 0.0001 | 0.1  |0.1| 0.3539|
> | 0.0001 | 0.01  |0.01| 0.3400|
> | 0.005 | 0.1  |0.1| 0.3482|
> | 0.005 | 0.01  |0.01| 0.3464|
> | 0.001 | 0.1  |0.1| 0.3371|
> | 0.001 | 0.01  |0.01| 0.3467|
>
>
>
>
>
> **Q4: How do you balance the training of the task network and augmentation network? For example, do you iteratively train either of the networks like GAN?**
>
> We alternately train G and F using the same batch of training data. When training the task network, we freeze G; and when training G, we freeze the task network.
>
>
>
> **Q3: As an important motivation, what if only one of the complementary modalities is augmented?**
>
> Thanks for the great question. Inspired by your comments, we have performed an additional set of experiments to investigate the effect of augmenting a single modality on Hateful Memes. In Hateful Memes, both text and image are required to decide if the content is hateful. Two modalities provide complementary information to each other. We run baseline augmentation only to one modality or independently to both modalities. With no augmentation, the multimodal network gives an accuracy of 0.6939. There are no consistent improvements. Essentially, performing augmentation naively to one modality or jointly without considering cross-modality relationships won’t lead to effective augmentation.
>
>
>
>
> |  | Image | Text | Image+Text|
> |--|--|--|--|
> | Trivial Augment |  0.7040 | 0.6860 | 0.7057|
> |MixUp| 0.6855|0.6777|0.6939|
> |Manifold Mixup| 0.6323 |0.7444|0.6878|
> |MixGen| 0.7427 | 0.6872 | 0.7510|

---

> > ### Comment · Reviewer_nNn5 · 2022-11-18
> > **Thanks for the rebuttal**
> >
> > Thanks for the detailed response. I confirm my major concerns are addressed. I have therefore improved my score.

---

### Official Review · Reviewer_cbnb · 2022-10-24

**Confidence:** 3
**Correctness:** 3
**Technical Novelty And Significance:** 3
**Empirical Novelty And Significance:** 3
**Recommendation:** 3

**Clarity, Quality, Novelty And Reproducibility:**

The idea is interesting. The writing seems to be good. However, the experiment is limited. The authors should consider the multimodal dataset with audio and visual modalities. And other multimodal data augmentation methods should be compared.

**Strength And Weaknesses:**

The idea is interesting. However, the experiment is limited. The authors should consider the multimodal dataset with audio and visual modalities. And other multimodal data augmentation methods should be compared.

**Summary Of The Paper:**

This paper proposes a Multimodal Data Augmentation model, which can augment multimodal data in feature space.

**Summary Of The Review:**

The idea is interesting.  However, the experiment is limited. The authors should do more experiments.

---

> ### Author Response · Authors · 2022-11-16
> **Response to Reviewer cbhb**
>
> We respectfully disagree. Our experiments are exhaustive, covering a range of multimodalities. In fact, we are even the first paper ever to consider a combination of image, text, and tabular data. In addition, we provide ablations to understand the effect of design decisions in our approach. Our experiments have been recognized as a strength by other reviewers.

---

### Official Review · Reviewer_huNS · 2022-10-24

**Confidence:** 4
**Correctness:** 3
**Technical Novelty And Significance:** 3
**Empirical Novelty And Significance:** 2
**Recommendation:** 8

**Clarity, Quality, Novelty And Reproducibility:**

Clarity:
The design choices such as using a VAE in the augmentation is not adequately discussed. While VAEs are effective, generative models such as diffusion models, GANs and normalizing flows could be suitable for the tasks. The reason for choosing VAEs should be elaborated.

Quality:
The paper is well-written and easy to follow and most of the claims are supported by experiments.

Novelty:
The aspect of experimenting with more than two modalities with the same network is novel, however, the framework is similar to prior work pointed to above.

Reproducaibility:
The work is not reproducible with the details provided in the paper. The manuscript mentions that the code will be released upon publication. It is important to provide necessary architectural details in the supplemental if not the main paper.

**Strength And Weaknesses:**

Strengths:
+ The method is applicable to any modality. The general framework proposed in this work can be applied across different input modalities. This is certainly a strong point of the work and most of the prior work on multimodal data augmentation caters to specific modalities.

+ Adequate experiments are performed to show the effectiveness of the approach. Experiments are performed for three modalities which is a positive. The performance improves across tasks and modalities. The usefulness of the consistency regularizer is also supported by ablations.

Weaknesses:
- The model formulation has limited novelty $f_{before}$ and is also present in the other related work [1,2,3,4,5] as pre-trained feature representations much like in this work. $\mathcal{G}$ is usually a VAE [1,2,3,5] encoding the latents and then the outputs of the decoder are used to minimize a certain loss (a reconstruction for consistency or a classifier in the latent space[1]). What are the crucial advantages of the current approach over these existing methods and how can be term the current approach as an “augmentation” approach since we do not modify the input data? In essence, can we use the above-mentioned approaches to sample many “z” to get the same effect as with the current approach?
- The architectural details are not provided. What kind of activations and VAEs are considered in work? What is the motivation/experimental reasoning behind using specific architectures? What is the design of the cross-attention module? This needs to be discussed in detail.
Support the following claims with an example:
 In related work: With early fusion, we capture the interaction between low-level features useful for multimodal tasks with strong cross-modal correlations.
The work in multimodal learning with late fusion is sparsely discussed. For example, the following work is relevant to the paper and should be adequately cited:
   1. Cross-Linked Variational Autoencoders for Generalized Zero-Shot Learning. Schönfeld et. al. CVPR 2019.
2. Latent normalizing flows for many-to-many cross-domain mappings. Mahajan et. al, ICLR 2020.
3. Learning two-branch neural networks for the image-text matching tasks. Wang et al, TPAMI 2019.
4. Learning robust visual semantic embeddings. Tsai et al, CVPR 2017.
5. Context object-split latent spaces for diverse image captioning.  Mahajan et al, NeurIPS 2020.
5 also introduces pseudo-captions for augmentation. How does it compare to the work proposed in the paper?


**Summary Of The Paper:**

This work identifies two critical challenges in multimodal data augmentation: First, Non-trivial augmentation for certain modalities such as numerical or categorical, and preservation of labels when the different modalities are augmented in isolation. This work proposes an easy-to-adapt learnable multimodal augmentation technique( LeMDA) that applies augmentation in the latent space. These augmentations are learned through adversarial training and semantic correspondence is ensured using consistency regularization.
Experiments on three modalities of images,  text, and tabular data show the approach's effectiveness and the improvement in accuracy with the proposed augmentation.

**Summary Of The Review:**

Overall, the paper introduces an approach for generalized multimodal data augmentation for any modality. The experiments are performed on various tasks and datasets where the method outperforms the prior state of the art. There are some concerns with the overlapping related work in vision and language (which can be addressed).

---

> ### Author Response · Authors · 2022-11-16
> **Response to Reviewer huNS**
>
>
> We really appreciate your supportive and thoughtful comments!
>
> **Q1: What are the crucial advantages of the current approach over these existing methods, and how can we term the current approach as an “augmentation” approach since we do not modify the input data? In essence, can we use the above-mentioned approaches to sample many “z” to get the same effect as with the current approach?**
>
> These mentioned works focus on generating latent features that are beneficial to a chosen task based on people’s intuition on what kind of feature representation would be beneficial for a specific task. For example, aligned image features and class embedding are beneficial for zero-shot settings. Because these methods are designed for a specific task, it is hard to generalize these methods to other tasks and other modalities.
>
> By contrast, we design our framework to be a general augmentation policy agnostic to modalities. Essentially, the augmentation policy is to automatically generate informative examples given the current task network. While other approaches to latent feature augmentations [e.g., 1, 2] have been shown to be effective for improving generalization, these approaches typically have been designed for single modality settings. We have recognized the particular value of latent feature augmentations for multiple modalities, modified the loss to provide informative examples while preserving semantic information, and are the first to consider a combination of tabular, text, and image data. Moreover, our approach achieves strong performance over a relatively wide variety of data types, is simple and automatic, and we have provided ablations to understand the importance of various design decisions.
>
> Given the relative scarcity of papers reasoning about multimodal data augmentation, and its fundamental significance, we believe our submission is providing an important and timely contribution — providing both a vision for how to approach this relatively new domain, as well as good results.
>
>
>
>
> [1]Varun Kumar, Hadrien Glaude, Cyprien de Lichy, and William Campbell. A closer look at feature space data augmentation for few-shot intent classification. CoRR, abs/1910.04176, 2019. URL http://arxiv.org/abs/1910.04176.
>
> [2]Xiaofeng Liu, Yang Zou, Lingsheng Kong, Zhihui Diao, Junliang Yan, Jun Wang, Site Li, Ping Jia, and Jane You. Data augmentation via latent space interpolation for image classification. ICPR, pp. 728–733, 2018. doi:10.1109/ICPR.2018.8545506
>
>
>
> **Q2: What kind of activations and VAEs are considered in this work? What is the motivation/experimental reasoning behind using specific architectures? What is the design of the cross-attention module?**
>
>
> We use ReLU activations. We consider two types of VAE as the augmentation network: MLP-VAE and Attention-VAE. In MLP-VAE, the encoder and decoder are 4 layer MLPs. In the Attention-VAE, the encoder and decoder are a self-attention module with 8 attention heads. The latent dimension in the VAE is set as 8. In this work, we use the VAE for its simplicity and efficiency. Our main focus is to demonstrate the effectiveness of a learnable augmentation framework for multimodal learning. We agree with you that other generative models could also be interesting to explore as part of our pipeline, in future work. But even with the simple VAE, we see clear performance benefits, which we view as an attractive feature of our approach.
>
>
> **Q3: Support the following claims with an example: In related work: With early fusion, we capture the interaction between low-level features useful for multimodal tasks with strong cross-modal correlations. The work in multimodal learning with late fusion is sparsely discussed.**
>
>
> Early fusion architectures can be designed to exploit the interaction between low-level features, making it a good choice for multimodal tasks with strong cross-modal correlations. For example, there exists a low-level correspondence in image captioning tasks because different words in the caption may relate to different objects in the image. We have also updated Section 2 with suggested references on late fusion.

---

> > ### Comment · Reviewer_huNS · 2022-12-01
> > **Thank you for the response**
> >
> > I have read the author's response and am satisfied with the clarifications. This approach would be helpful in multimodal methods where data augmentation is challenging. I would therefore keep my score

---

### Official Review · Reviewer_PrKb · 2022-10-25

**Confidence:** 4
**Correctness:** 3
**Technical Novelty And Significance:** 4
**Empirical Novelty And Significance:** 4
**Recommendation:** 6

**Clarity, Quality, Novelty And Reproducibility:**

Clarity: There are some questions that I posted on the Weakness session. But overall the idea and logic is clear to me.
Quality: The written is in good quality and the figures/tables/experiments are well-designed.
Novelty: I think the proposed meothod is a novel idea.
Reproducibility: The manuscript provided some implementation details and promised to open-source code upon publication.

**Strength And Weaknesses:**

# Strength
The paper is well-written and easy to follow. The experimental results are strong and the improvements from the proposed method are significant. The ablation studies are well-designed and clearly illustrated the effectiveness of the design choices.

# Weakness
* I wonder if there are typos on the gradients for updating augmentation network G. Should the gradient be written as -\partial L(y^hat_G) + \partial L_{consistent} instead? Updating network G with the gradient -\partial L(y^hat) doesn't make sense to me as i believe \partial L(y^hat) \partial G is zero? (this notation consistently appears in Fig. 2, Alg. 1 and main text).
* I feel one table illustrating the training cost (FLOPS/Params/CPU wall time) when using different augmentation is necessary to better understand the proposed augmentation strategy.
* Is the update step on augmentation network G and task network F happening within one iteration (using the same batch of training data) or the updating steps between G and F are alternating as in GAN training?
* When updating these two networks, is there any hyperparameter that need to be adjusted (e.g. the weight between two losses). If there is any augmentation specific hyperparameter, is it consistently used across different experiments/datasets?
* The overall improvements brought by the proposed augmentation across different datasets have large variance (from 0.4% - 6%). It would be better if the authors provides more discussions and insights on why the proposed method works better on certain dataset than others.

**Summary Of The Paper:**

This manuscript proposed a new data augmentation strategy applied on the feature space for multi-modal classification task. Specifically, the authors applied learnable augmentation network, in the form of VAE, to perturb the encoded embeddings from different modalities. Experiments are conducted on 8 multimodal datasets and compared with the baseline, the proposed augmentation strategy significantly improves the classification results (up to 6%).

**Summary Of The Review:**

Overall I think this manuscript is a good paper. I give marginal accept for now and would like to hear authors' feedbacks on my comments in the weakness session.

---

> ### Author Response · Authors · 2022-11-16
> **Response to Reviewer PrKb**
>
> Thank you for your helpful and supportive comments! We have tried to carefully address your questions. Please also see our general comment to all reviewers. We hope you can consider raising your score in light of our response, which includes several results inspired by your questions, and the timeliness and significance of this work. Multimodal data augmentation is at a very nascent stage of development in the community, and we believe this paper could have quite an impact if given the opportunity.
>
>
> **Q1: I wonder if there are typos on the gradients for updating G.**
>
> Great catch. The gradient should be written as -\partial L(y^hat_G) + \partial L_{consistent}. We have updated this notation throughout our manuscript. Thanks for pointing it out.
>
>
> **Q2: Illustrating the training cost when using different augmentation is necessary to understand the proposed augmentation strategy better.**
>
> Since LeMDA involves training an augmentation network trained in addition to a task network, LeMDA does incur some additional expense. We summarize the training throughput, measured in it/second in the following table. Generally, we observe lower throughput for LeMDA.
>
> While the training costs are relatively manageable, they could be improved by reducing the frequency of updating the augmentation network. Currently, the augmentation network is updated in every iteration, though the parameters for the task network change relatively slowly, especially in the later stages of training. This could be an interesting direction for future work.
>
>
> |  | Multimodal Network | Input Augmentation | MixUp| Manifold MixUp| MixGen|LeMDA|
> |--|--|--|--|--|--|--|
> | Hateful memes | 2.39  | 2.17 | 2.35| 1.63| 2.35| 1.41|
> | Food101 | 4.27  | 4.46 | 4.31|4.48| 4.47| 2.21|
> | Petfinder |2.36  | 2.29 | 2.95|2.36| - | 1.87|
> | Melbourne Airbnb | 5.66|5.94 | 5.59|5.69| - | 4.13|
> | News Channel |8.14  | 7.18 | 7.31|7.12| - | 5.12|
> | Wine Review | 12.54  | 11.60 | 11.89|11.46| - | 6.28|
> | Wine Review | 12.37  | 12.57 | 12.62|12.21| - | 6.69|
>
>
>
> **Q3: Is the update step on augmentation network G and task network F happening within one iteration (using the same batch of training data), or are the updating steps between G and F alternating as in GAN training?**
>
> We use the same batch of training data and perform two separate passes over F for gradient calculations. In theory, only one pass is needed, though our implementation is more convenient when used in conjunction with PyTorch autograd.
>
>
> **Q4: Is there any hyperparameter when training two networks? Is it consistent across datasets?**
>
>
> Since the optimization for the augmentation network is a min-max game, there are hyperparameters to balance the loss. Specifically, w_1 \partial L(y^hat_G) + w_2\partial L_{consistent} +w_3 \partial L_{KL}, where \partial L_{KL} is the VAE KL regularizer.
>
> We emphasize that, relatively speaking, our method does not involve many hyperparameters and that performance is not particularly sensitive to the values of these hyperparameters. For example, in the following table, we summarize the hyperparameters we use for Table 1. The hyperparameters are pretty consistent across datasets.
>
>
> |  | w1 | w2| w3|
> |--|--|--|--|
> | Hateful Memes | 0.0001 | 0.1 |0.1 |
> | Food101| 0.0001 | 0.1 |0.1 |
> | Petfinder | 0.0001 | 0.1 |0.1 |
> | Melbourn Airbnb | 0.001 | 0.1 |0.1 |
> | New Channel | 0.0001 | 0.1 |0.1 |
> | Wine Review | 0.0001 | 0.1 |0.1 |
> | Kick Starter FUnding | 0.0001 | 0.1 |0.1 |
> | SNLI-VE | 0.001 | 0.1 |0.1 |
>
>
> Additionally, inspired by your comments, we investigate the influence of different combinations of hyperparameters and summarize the accuracy in the following table on Petfinder. We observe a consistent improvement  over multimodal network across various combinations.
>
> | Multimodel Network | | | 0.2911|
> |--|--|--|--|
> | w1 | w2 | w3| Accuracy|
> | 0.0001 | 0.1  |0.1| 0.3539|
> | 0.0001 | 0.01  |0.01| 0.3400|
> | 0.005 | 0.1  |0.1| 0.3482|
> | 0.005 | 0.01  |0.01| 0.3464|
> | 0.001 | 0.1  |0.1| 0.3371|
> | 0.001 | 0.01  |0.01| 0.3467|

---

> > ### Author Response · Authors · 2022-11-16
> > **Response to Reviewer PrKb**
> >
> > **Q5: Discussion and insights on why certain datasets have much higher improvements.**
> >
> >
> > Thanks for the thoughtful question!
> >
> > We believe the size of the data is a contributing factor to where LeMDA performs best. Combining Table 1 and Table 2, we observe that LeMDA’s improvements are particularly significant on two smaller datasets (Hateful Memes, Perfinder), where our augmentation approach is compensating for lack of information that is helpful for learning a good representation — and, relative to other augmentation procedures, is able to provide information across correlated modalities.
> >
> > Indeed, LeMDA is most valuable relative to alternatives when it is not a priori clear what transformations the data should be invariant to. This advantage particularly shines when a combination of tabular and text data are the input modalities, such as in wine reviews. In these modalities it is much less clear what transformations we may wish to respect relative to images (where the prediction task may be invariant to translations and rotations, for example). The performance gains are also significant when there are complex interactions between the different data modalities. In addition to their smaller size, both Petfinder, which has image+tabular+text modalities, and hateful memes, where standard image transformations could render the textual descriptions benign, have such structure.
> >
> > We appreciate the question and will highlight these observations in the final version. We have generally made an effort to provide interpretability, for example in our conceptual explanation of consistency regularization, and in our ablation of design decisions.

---

> > > ### Comment · Reviewer_PrKb · 2022-12-03
> > > **Thanks**
> > >
> > > Thanks the authors for their reponses - they answered most of my questions and I'd keep my rating.

---

### Author Response · Authors · 2022-11-16
**Response to all Reviewers and AC**



We thank reviewers for thoughtful and highly supportive feedback! We want to emphasize that our paper is making a timely and substantial contribution. While biological learning agents strongly leverage information from multiple modalities, machine learning approaches have comparatively been focused almost exclusively on single-modality tasks. Multimodal data augmentation is even more rarely explored, despite its extraordinary impact in the single modality setting.

Our approach to multimodal data augmentation is simple, automatic, and empirically effective. Moreover, we provide exhaustive empirical investigations, showing improved performance over specialized augmentations, including augmentations tailored to vision+text, and the first ever results on images + text + tabular data. It is not typical for a paper to provide both state-of-the-art results in addition to results on entirely new types of data.

Inspired by reviewer comments, we have additionally updated the manuscript to incorporate several updates. We summarize the major changes as follows:


 1. Architecture details of the Augmentation Network in Appendix A.1 and Section 3.3.
 2. Detailed training implementation in Appendix A.2.
 3. Additional experiment and discussion on the training cost in Appendix B.1.
 4. Additional experiment and discussion on the hyperparameters choice to balance losses on the Augmentation Network and their influence in Appendix B.2.
 6. Observations when only one of the complementary modalities is augmented in Section 4.5.
 7. Updated Section 2 with more references.


We hope that reviewers will consider updating their scores in consideration of these updates, and the significance and timeliness of the work.

---

### Decision · Program_Chairs · 2023-01-20

**Decision:**

Accept: poster

**Justification For Why Not Higher Score:**

The contribution is sound and the experimental evidence extensive. Some choices could be better motivated - e.g. the use of a VAE instead of other generative models in this context. The authors leave this question as future work.

**Justification For Why Not Lower Score:**

Relevant topic, sound contribution, convincing experimental evidence to support the claims.

**Metareview: Summary, Strengths And Weaknesses:**

This paper was reviewed by four knowledgeable referees. Overall, the reviewers found the paper timely, the contribution interesting, and the evaluation rather extensive. Reviewers raised concerns about the missing discussion w.r.t. training costs (PrKb), the variance in the results across datasets (PrKb), the justification w.r.t. some design and hyper-parameter choices (PrKb, huNS, nNn5), the advantages over some existing methods (huNS), and the novelty (nNn5). The authors engaged with the reviewers during the discussion phase and adequately addressed most of their concerns. The authors ask the AC to disregard the review by cbnb. The AC goes over the review and agrees with the authors that the review is unsubstantial and does not provide detailed actionable feedback. The AC agrees with the other reviewers' assessment that the contribution is timely and well executed, and could stimulate future research. Therefore, the AC recommends to accept.

**Note From Pc:**

if the above contains the word "oral" or "spotlight" please see: "oral" presentation means -> notable-top-5% and "spotlight" means -> notable-top-25%. As stated in our emails, we are disassociating presentation type from AC recommendations